# Multimodal Manifold Learning for Clonally Constrained Trajectory Inference

**Irene Bonafonte-Pardàs** *
Institute of Computational Biology
Helmholtz Center Munich
Neuherberg, Germany

**Myriam Lizotte** *
Mila – Quebec AI Institute
Montréal, Canada
Department of Mathematics and Statistics
Université de Montréal

**Guy Wolf** †
Mila – Quebec AI Institute
Montréal, Canada
Department of Mathematics and Statistics
Université de Montréal

**Benjamin Schubert** †
Institute of Computational Biology
Helmholtz Center Munich
Neuherberg, Germany

## Abstract

A central goal of single-cell transcriptomics is to reconstruct dynamic cellular processes from static scRNA-seq snapshots, yet most trajectory inference methods rely on transcriptomic similarity as a proxy for developmental linkage — an assumption that frequently fails. While lineage tracing overcomes this limitation, it requires genetic perturbations and specialized longitudinal designs. In adaptive immune cells, T and B cell receptors (AIRs) naturally encode clonal ancestry and are routinely sequenced alongside the transcriptome, providing lineage information in standard snapshot datasets, but existing trajectory methods are not adapted to exploit this signal. Here, we lay the foundation for incorporating AIR-encoded lineage information into trajectory inference by biasing RNA-based diffusion maps toward AIR-consistent paths, thereby integrating lineage constraints into learned cell-state representations. Across simulations of increasing complexity, our multimodal approach recovers more biologically plausible trajectories than RNA-only baselines. While optimized for lymphocyte differentiation, the framework generalizes to other endogenous lineage barcodes, such as mitochondrial mutations.

## 1 Introduction

Understanding how cells differentiate from progenitor states into mature end states is a central problem in developmental biology and underlies the progression of many diseases. Single-cell RNA sequencing (scRNA-seq) has enabled the high-resolution profiling of gene expression at the level of individual cells, making it possible to computationally reconstruct dynamic biological processes from static measurements, a task known as trajectory inference (TI) Wagner & Klein (2020). In a scRNA-seq experiment of a developing system, cells are collected at a single time point but cover the continuum of the process. TI methods aim to disentangle the trajectories taken by cells and to order individual cells along these trajectories, assigning each cell a *pseudotime* and a *fate* that reflect its progression and commitment from an initial to a terminal state.

Since the introduction of pseudotime by Trapnell et al. (2014), a large body of work has been proposed for trajectory inference based solely on transcriptomic similarity, including Wanderlust (Bendall et al., 2014), Diffusion Pseudotime (DPT) (Haghverdi et al., 2016), Slingshot (Street et al., 2018), PAGA (Wolf et al., 2019), PHATE Moon et al. (2019), Palantir Setty et al. (2019), and CellRank Weiler et al. (2024), each focused on a specific subproblem, but with substantial methodological overlap. These and many subsequent approaches (Saelens et al., 2019) typically assume that

---

*These authors contributed equally to this work.
†These authors jointly supervised this work.

cells close in gene expression space are also close in developmental time, and that cells are densely sampled along pseudotime. However, these assumptions often break down: distinct lineages may converge to similar transcriptional end states, such as exhaustion, or diverge rapidly despite transient transcriptional similarity Tritschler et al. (2019). As a result, trajectory inference based on RNA alone can yield biologically implausible paths.

A promising strategy to address these limitations is to incorporate side information that reflects true developmental relatedness. *Prospective lineage tracing* provides such information through the incorporation of genetic barcodes into ancestor cells to track their descendants (Kester & Van Oudenaarden, 2018). Several computational methods combine such data with transcriptomic information to infer ancestor-descendant couplings, including CoSpar (Wang et al., 2022), scTrace+ (Guo et al., 2025), and PORCELAN (Schlüter & Uhler, 2025), as well as optimal transport–based approaches such as Waddington-OT (Schiebinger et al., 2019), superOT (Prasad et al., 2020), and Moslin (Lange et al., 2024). To enable lineage tracing in a wild-type setting *in vivo*, *retrospective lineage tracing* exploits naturally occurring genetic mutations, such as copy number variations or mitochondrial variants Ludwig et al. (2019). However, in this setting, clonal labels are often sparse and noisy, and computational analyses are typically restricted to the reconstruction of clonal phylogenetic trees Jones et al. (2020), cell type level phylogenies Sashittal et al. (2025), or the use of clonal-label derived features to improve velocity fields estimation, without directly modeling clonal relationships Wang et al. (2024).

Adaptive Immune Receptors (AIR) of B and T lymphocytes can also be viewed as retrospective lineage barcodes. Each lymphocyte expresses a unique AIR, generated through highly combinatorial genetic recombination (Pai & Satpathy, 2021), making it unlikely that two lymphocytes sharing the same receptor sequence do not descend from a common ancestor. AIRs are routinely sequenced alongside the transcriptome in standard scRNA-seq experiments with high accuracy. As such, AIR data provides high-confidence clonal information that is readily available in snapshot datasets, without genetic perturbation or specialized experimental design.

Only a small number of approaches have explored the use of AIR information to learning representations suitable for pseudotime and fate inference, including Dandelion (Suo et al., 2024), which exploits a bias in AIR-gene usage composition during early T-cell development in the thymus, but is not applicable to other scenarios, and Xie et al. (2022), which characterizes clones based on their preferred transcriptomic-based trajectories, but does not use AIR to refine trajectories. mvTCR (Drost et al., 2024), and CoNGA (Schattgen et al., 2022) learn joint transcriptomic and AIR representations, but are optimized for antigen specificity prediction, pushing together clonally related cells, instead of highlighting intra-clonal evolution.

In this work, we introduce **T-time**, a principled framework for trajectory inference in snapshot scRNA-seq data, integrating endogenous lineage information with transcriptomic similarity. We propose to use optimal transport-based clonal similarities to impose lineage constraints on diffusion-based representations, enabling lineage-consistent trajectory inference without longitudinal sampling. We define a clonally-constrained transition map to perform (**i**) cellular representation learning, (**ii**) terminal fate prediction, and (**iii**) pseudotime inference within a unified framework. To prevent clonal collapse, we use a step-size controlled alternating diffusion variant that balances modalities while preserving local neighborhood information. Using synthetic data, we confirm that RNA-only approaches fail under transcriptomic convergence and in the absence of intermediate cell states, and that these failures can be resolved by incorporating clonal information. While motivated by T-cell differentiation, our framework generalizes to any endogenous lineage barcode and provides a general strategy for constraining representation learning with lineage information.

## 2 LINEAGE-CONSTRAINED TRAJECTORY INFERENCE

### 2.1 PROBLEM FORMULATION

Let $\mathcal{X} = \{x_i\}_{1 \leq i \leq n} \subset \mathbb{R}^{n \times N}$ denote a dataset of $n$ single cells, where each sample $x_i$ is an $N$-dimensional gene expression vector. Each cell is additionally associated with an endogenous lineage barcode, here an adaptive immune receptor (AIR), encoding clonal ancestry. Let $\mathcal{C} = \{c_k\}_{1 \leq k \leq n_{\text{clones}}}$ denote the set of observed clonotypes, and let $C : \mathcal{X} \to \mathcal{C}$ map each cell to its clonotype.

We seek a multimodal transition matrix $P$ yielding 3 outputs: a low-dimensional cell-state representation $Z = \{z_i\}_{i=1}^n \subset \mathbb{R}^{n \times d}$, with $d \ll N$ of $\mathcal{X}$, that reflects progression along an underlying branching biological process; a *pseudotime* function $\tau : \mathcal{X} \to \mathbb{R}^+$ to map each cell to a scalar indicating its position along this process, and a *terminal branch* probability function $\mathbb{P}$, reflecting the likelihood of commitment to each fate. $Z$ should have the following properties: **(i)** local structure in $Z$ preserves transcriptomic similarity, **(ii)** inferred trajectories are consistent with clonal similarity, and **(iii)** cells within the same clone form continuous paths rather than compact clusters.

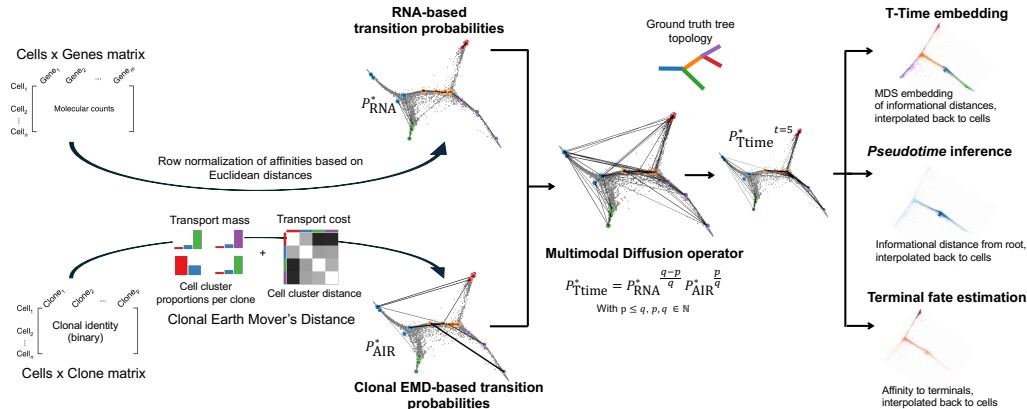

Figure 1: **Graphical depiction of the T-time algorithm.** The input space is first reduced via landmark representatives, on which transcriptomic- and AIR-based diffusion operators are defined. These modality-specific operators are fused using alternating diffusion with modality-dependent step sizes. The resulting multimodal diffusion operator is exponentiated to $t$, smoothing transitions and emphasizing dominant trajectories to yield a clonally constrained diffusion map for visualization, pseudotime inference, and fate estimation.

## 2.2 FUSING MODALITIES VIA STEP-SIZE-CONTROLLED ALTERNATING DIFFUSION

An overview of our proposed approach is shown in Fig. 1. We treat gene expression and AIR as two different views or modalities of the same system and construct separate diffusion processes on each. In detail, for each modality $i$ in $\{\text{RNA}, \text{AIR}\}$, we construct a diffusion operator $P_i$. $P_{\text{RNA}}$ is defined by row-normalizing the affinity matrix $W_{\text{RNA}}$ which is made by applying the $\alpha$-decay kernel (Moon et al., 2019) on pairwise Euclidean distances. $P_{\text{AIR}}$ is defined in detail in section A.1.1. Next, we combine the two views in the following way:

$$P_{\text{T-time}} = P_{\text{RNA}}{}^{\frac{q-p}{q}} P_{\text{AIR}}{}^{\frac{p}{q}}$$

for some $p \leq q$ with $p, q \in \mathbb{N}$. Conceptually, we can interpret $(P_{\text{RNA}})_{i,j}$ and $(P_{\text{AIR}})_{i,j}$ as the probabilities of transitioning from cell $x_i$ to cell $x_j$ in a one-step random walk, where transitions are more likely between cells that are similar in the given modality. Then, we can view $(P_{\text{T-time}})^q$ as encoding the transition probabilities of a $q$-step random walk, where the first $q - p$ steps follow the RNA transitions and the next $p$ steps the AIR transitions. The parameter $p$ controls the relative importance of the views in the overall operator. We set $p \ll q$ to allow our random walk to take several steps in $P_{\text{RNA}}$ before requiring it to reach a cell with the same clonal identity, which helps deal with clonal sparsity. We name our method T-time $p/q$ to indicate the selected view balance, where T-time alone refers to T-time 1/5, our preferred ratio.

Taking the $q$th root, as initially proposed by Mendelman & Talmon (2025), ensures that the resulting multimodal diffusion operator overall still represents single-step transition probabilities. This mitigates oversmoothing when exponentiating $P$ for downstream tasks. We treat the computation of a stochastic $q$-th root of an $m \times m$ transition matrix $P$ as a constrained optimization problem:

$$\min_{X \in \mathbb{R}^{m \times m}} \|X^q - P\|_F^2 \quad \text{subject to} \quad X\mathbf{1} = \mathbf{1}, \ X \geq 0 \text{ with } \mathbf{1}_i = 1 \ \forall 1 \leq i \leq m.$$

solved through a projected gradient method optimized with ADAM updates, and stochasticity enforced at each step through Quasi Optimization of the root Matrix (Kreinin & Sidelnikova, 2001).

**Defining clonal distances.** We compute distances between clones based on optimal transport (OT) (Peyré & Cuturi, 2018). For computational efficiency, we first partition the dataset into $m' = n/10$ clusters and represent each clone as a distribution over these clusters. We then compute the optimal transport cost, i.e., earth mover's distance (EMD), between each pair of clones, using informational distance (Moon et al., 2019) as the ground cost so that the matchings better reflect the latent manifold. This yields an $n_{\text{clones}} \times n_{\text{clones}}$ matrix $D_{\text{AIR}}$ of clonal distances. See section A.1.1 for details.

**Landmarking for computational efficiency**: We use a landmarking approach as in Moon et al. (2019) to identify $m = n/5$ landmarks via spectral clustering and build $P^*_{\text{RNA}}$, an $m \times m$ matrix of transition probabilities between landmarks. This algorithm also provides an $n \times m$ cell-to-landmark transition matrix $P^{\text{cell-landmark}}_{\text{RNA}}$ that is needed to interpolate back from landmarks to individual cells. Using the same set of RNA-based landmarks, we can define an $m \times m$ matrix $P^*_{\text{AIR}}$ by running a second OT step viewing the landmarks as distributions over clones, with $D_{\text{AIR}}$ as the OT cost. We also consider an alternative approach based on Jaccard similarity (Appendix A.1.2). The landmarked T-time modality fusion thus becomes:

$$P^*_{\text{T-time}} = P^*_{\text{RNA}}{}^{\frac{q-p}{q}} P^*_{\text{AIR}}{}^{\frac{p}{q}}$$

### 2.3 LINEAGE-CONSTRAINED STRUCTURE VISUALIZATION AND TERMINAL FATE INFERENCE

Our method yields a diffusion operator $P_{\text{T-time}}$, but we also consider other constructions $(P_{\text{AD}}, P_{\text{ID}}, P_{\text{RF-PHATE}})$ based on existing methods that define a multi-modal transition matrix, which are described in Appendix A.1.3. Starting with any diffusion operator $P \in \{P_{\text{T-time}}, P_{\text{AD}}, P_{\text{ID}}, P_{\text{RF-PHATE}}\}$, we raise it to the power of $t$, selected via entropy following Moon et al. (2019) to promote dominant trajectories in the data. We then define the diffusion potential $Q = -\log P^t$, and the informational distance matrix (Moon et al., 2019): $D^2_{i,j} = \sum_{k=1}^n (Q_{i,k} - Q_{j,k})^2$. We apply multidimensional scaling (MDS) to $D$ to obtain the 2D embeddings used for visualization.

For pseudotime and fate inference, we adjust user-provided root $x_r$ and terminal states $x_{e_i}$ to align with the limits of the learned manifold. Pseudotime is computed as informational distance from the root cell: $\tau(x_i) = f(D_{i,r})$ where $f$ is a min-max normalization function. For fate inference, we use the diffusion potential to terminal cells, passed through a Gaussian Kernel $K$ to create similarities, normalized to get a probability distribution over the $n_{\text{end}}$ terminal fates: $\mathbb{P}(\text{Fate of } x_i = x_{e_j}) = f'(K(Q_{i,e_j}))$ where $f'$ is a softmax function adjusted to fit the observed frequencies. Finally, since the diffusion operator is at the landmark level, we extend the embeddings, pseudotimes and fates to all cells by multiplying by the cell-to-landmark transitions $P^{\text{cell-landmark}}_{\text{RNA}}$.

### 2.4 EXPERIMENTAL SETUP

To assess the benefits of incorporating AIR information into trajectory inference, we make use of synthetic data generated with a clonal extension of PROSSTT Papadopoulos et al. (2019) to simulate biologically plausible differentiation processes while having access to the ground-truth tree topology, cell-level fates, and pseudotime, across controlled transcriptomic noise and clonal distributions (see Appendix A.1.4). This setup helps to expose failure modes of RNA-only methods when transcriptomic similarity does not reflect developmental linkage, and allows us to quantify the improvement induced by clonal information. Results in this controlled synthetic setting are intended as a proof of concept, rather than a comprehensive empirical evaluation.

**Metrics**: We evaluate T-time cellular representations, terminal fate assignment, and pseudotime inference qualitatively and quantitatively using six different metrics. We measure structure preservation in the learned cellular embeddings with the *Mantel test* for global structure and *trustworthiness* for local neighborhood preservation (k= 50). Here, ground-truth cell similarity is defined as a Gaussian kernel applied to Euclidean distances in the noise-free gene program space, multiplied by a binary mask indicating whether cells are in adjacent branches in the lineage-restricted tree. Terminal fate predictions are evaluated at the branch level, as *total variation* (TV) in the predicted distribution of cells across terminal branches; and at the cell level, as the *mean accuracy* of the cell-to-terminal branch assignments. We use global and within-branch inferred and ground-truth pseudotime *correlation* to evaluate the ordering of cells at the global and local level.

**RNA-Baselines**: We compare T-time to well-established RNA-based methods, including: diffusion maps (Coifman & Lafon, 2006), PHATE (Moon et al., 2019), and UMAP (Mcinnes et al., 2020)

for cell representation learning; diffusion pseudotime (Haghverdi et al., 2016) and Palantir (Setty et al., 2019) for pseudotime inference; and Palantir (Setty et al., 2019) and CellRank2 (Weiler et al., 2024) for fate prediction. On all tasks, we also compare to the T-time model in the absence of clonal information (T-time 0/5) to isolate the contribution of AIR-derived signals.

# 3 RESULTS

## 3.1 T-TIME RESCUES TRAJECTORY INFERENCE FOR CONVERGING CELL STATES

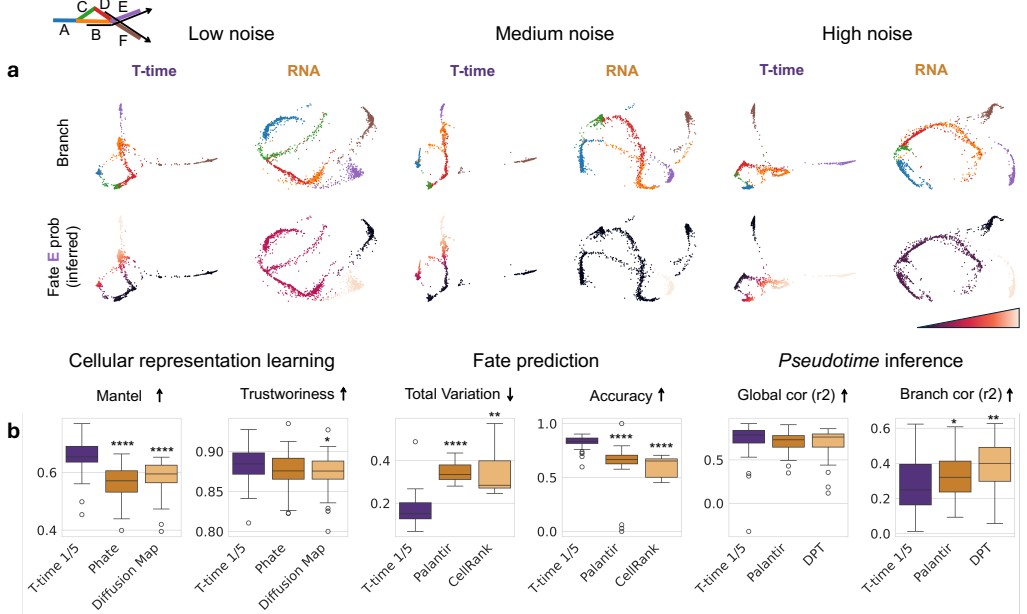

Figure 2: **T-time comparison to RNA-based models. (a)** Representative T-time and Phate embeddings of a cross-converging tree across three levels of noise, colored by branch and inferred probability to commit to fate E. For RNA-based models, the results with the best-performing method are shown in each case. **(b)** T-time quantitative comparison to RNA-based models on cellular representation learning, fate prediction, and pseudotime inference tasks. Stars indicate significance in a paired, two-sided t-test compared to T-time 1/5. *: $p < 0.05$, **: $p < 0.01$, ***: $p < 0.001$, ****: $p < 0.0001$. Arrows indicate the desired direction of each metric. Boxplots summarize metric distributions across the 43 experiments (median, interquartile range, and outliers).

We benchmarked T-time against RNA-based models using an immunologically motivated T cell differentiation topology. In this scenario, naïve T cells activate upon antigen encounter, differentiate into effector cells, converge into a memory state, and subsequently re-differentiate upon secondary antigen exposure. This process generates a tree-like structure with branching events that converge to a fixed point, from which multiple branches emerge again, representing memory reactivation. For this comparison, we simulated 43 datasets (average 1,908 cells, 200 clones) with progressively increasing levels of transcriptomic noise.

Across noise levels, T-time reliably recovered the underlying tree topology, while RNA-based embeddings failed to correctly connect the branches, as illustrated in representative embeddings (Fig. 2a). For example, branch A should be directly connected to its descendant B, and similarly for D and F. In the low noise scenario, the RNA embedding shows distance between these pairs, while the T-time embedding creates clearer connections between them.

This structural advantage was reflected in a significantly higher Mantel score for T-time ($0.653\pm0.063$ vs $0.582\pm0.059$ for diffusion map; paired T test, $p= 2e^{-5}$), without compromising local neighborhood preservation ($0.884\pm0.023$ vs $0.876\pm0.024$ for Phate, $p= 0.24$, Fig 2b). In line with this, T-time consistently disentangled terminal fates at converging branches *B* and *D*. Cells in

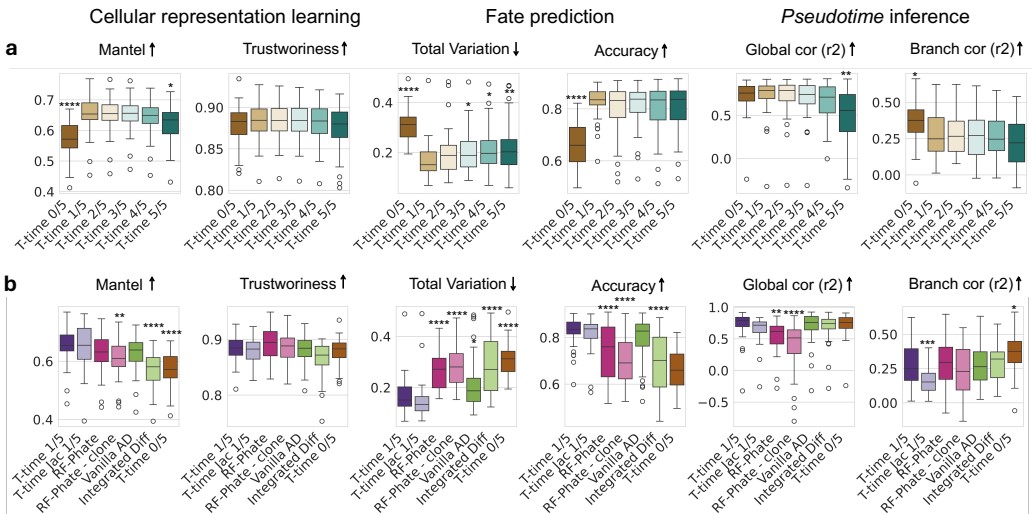

Figure 3: **Comparison of clonally constrained trajectory inference approaches**. Quantitative evaluation of T-time compared to **(a)** T-time with different modality balancing and **(b)** alternative approaches to multimodal transition maps, on cellular representation learning, fate prediction, and pseudotime inference tasks. Stars indicate significance in a paired, two-sided t-test compared to T-time 1/5. *: $p < 0.05$, **: $p < 0.01$, ***: $p < 0.001$, ****: $p < 0.0001$. Arrows indicate the desired direction of each metric. Boxplots summarize metric distributions across the 43 experiments (median, interquartile range, and outliers).

branch *B* were assigned a high terminal probability to *fate E*, while cells in branch *D* were correctly biased to *fate F* (Fig. 2a). This resulted in a significantly lower average fate TV of 0.170±0.071, compared to Palantir ($0.343 \pm 0.042$, $p= 3e^{-16}$) and CellRank (0.350±0.117, $p= 0.005$), as well as a significant improvement in per-cell terminal fate prediction accuracy (T-time 0.825±0.062, Palantir - best RNA-model, 0.639±0.196; $p= 2e^{-6}$, Fig. 2b).

In contrast, improved structural recovery conferred no advantage for pseudotime inference (SFig. 1a). Global pseudotime correlation were comparable to RNA-based methods (T-time: 0.725±0.216, DPT - best RNA-model: 0.698±0.169), while within-branch correlation was significantly lower (T-time: 0.276±0.142, DPT - best RNA-model: 0.386±0.141; $p$=0.002). At very high transcriptomic noise levels, T-time also showed no quantitative gain in downstream tasks, potentially due to noise propagation to transcriptionally defined landmarks and clonal distances used in $P_{AIR}^*$ (SFig. 1b).

Together, these results highlight the value of integrating clonal information to recover tree topologies that are not discernible from transcriptomic data alone, as reflected both in visual representations and improved terminal fate inference. Although T-time embeddings maintained a high trustworthiness (0.884±0.023), indicating preservation of local structure, the decrease in within-branch pseudotime correlation suggests that some local transcriptomic information is lost through integration with clonal data, underscoring the importance of carefully balancing the influence of both modalities.

## 3.2 FINE-GRAINED MODALITY BALANCE IS NECESSARY FOR FAITHFUL RECONSTRUCTION

To further explore the impact of RNA to AIR balance, we used the same experimental setup to compare T-time's performance across different $P_{RNA}^*$ and $P_{AIR}^*$ exponents (Fig. 3a). Expanding the previous findings, the RNA-pure version *T-time 0/5* significantly underperformed in fate prediction tasks (e.g. fate TV T-time 0/5: 0.310±0.067, 1/5: 0.170±0.071; $p$=1$e^{-11}$) and global structure preservation (Mantel T-time 0/5: 0.573±0.058, 1/5: 0.653± 0.063; $p$=1$e^{-6}$), while the AIR-pure version *T-time 5/5* significantly worsened global pseudotime inference (5/5: 0.493±0.345, 1/5: 0.725±0.216; $p$=0.003). Across all tasks, except within-branch pseudotime correlation, we observed a strong, consistent performance increase when adding a small amount of clonal information (i.e., from *T-time 0/5* to *T-time 1/5*), followed by a gradual decline as the clonal exponent increased. This pattern was most apparent for fate prediction tasks, where T-time achieved the largest improvements.

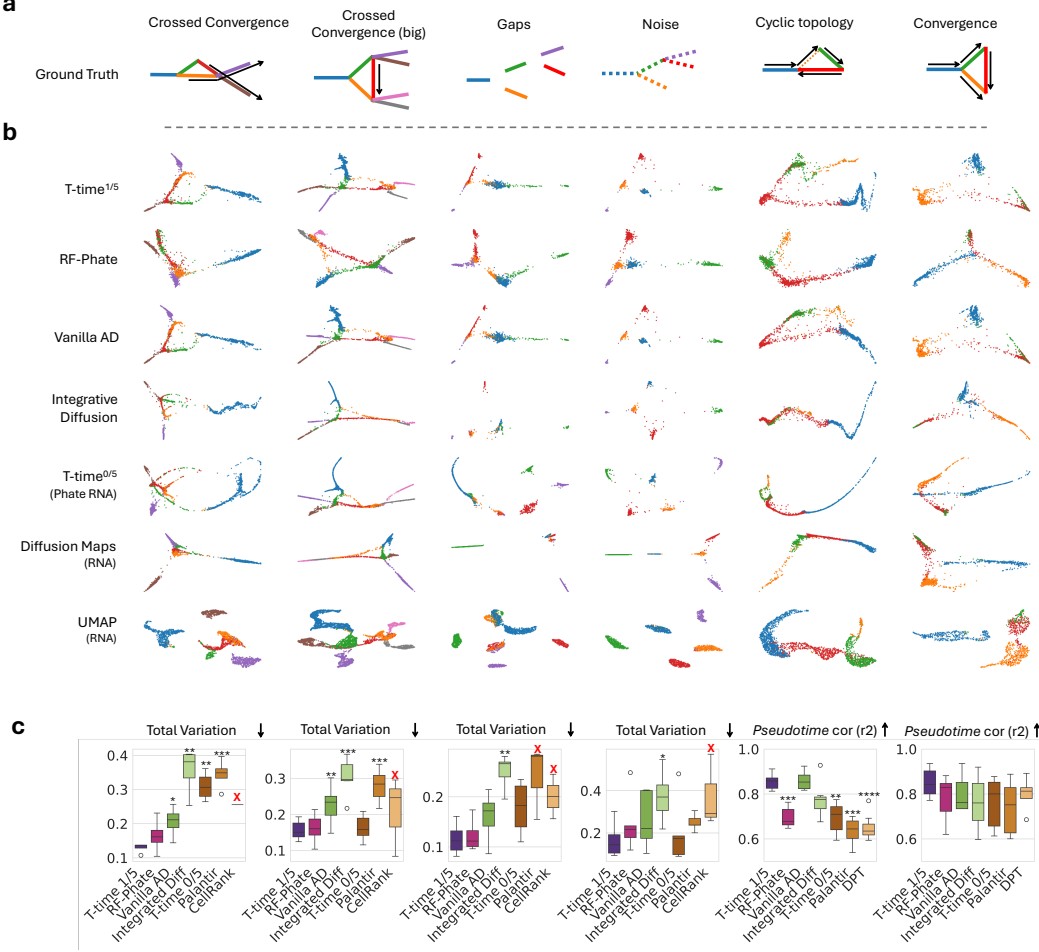

Figure 4: **T-time recovers the underlying tree structure across biologically inspired scenarios**. **(a)** Drawings indicating the ground truth tree topology. Color indicates branch, arrows indicate directionality for converging branches. **(b)** Representative cellular embeddings per dataset, comparing T-time to standard RNA-based embeddings, colored by branch. **(c)** Quantitative comparison of T-time to RNA-based methods and alternative methods to multimodal transition maps on tasks selected to represent the expected RNA failure mode on each dataset. Stars indicate significance in a paired, two-sided t-test compared to T-time 1/5. *: $p < 0.05$, **: $p < 0.01$, ***: $p < 0.001$, ****: $p < 0.0001$. Red cross indicates that the experiments failed in more than $40\%$ of the datasets. Arrows indicate the desired direction of each metric. Boxplots summarize metric distributions across experiments (median, interquartile range, and outliers).

Next, we investigated alternative integration approaches, both unsupervised and supervised (Fig. 3b; Appendix A.1.3) on the same experimental setup. To fully evaluate the importance of fine-grained modality balance, we compared to Alternating Diffusion (AD), which combines modalities with a fixed one-to-one ratio. Although this approach generally obtained good results, it significantly underperformed T-time in fate prediction (AD: 0.212±0.104, T-time 1/5: 0.170±0.071; $p$=0.02). Integrated Diffusion (ID) balances modalities by independently optimizing the transition matrix step size $t$ for each modality prior to integration. While shown to be useful for RNA-seq and ATAC-seq integration (Kuchroo et al., 2021), it significantly underperformed across the majority of metrics in our experiments (Fig. 3b). This behaviour is most likely explained by the extremely low entropy of $P_{AIR}^*$, which rapidly collapses with increasing $t$, compared to $P_{RNA}^*$ (SFig. 2a). To reach the optimal RNA to AIR balance, the total number of steps is too high for the entropy of our dataset, and $P_{ID}$ collapses to a single dominating diffusion component (SFig. 2b).

As a supervised alternative to diffusion-based multimodal integration, we explored RF-PHATE. Consistent with its objective, RF-PHATE generated good cellular embeddings, both qualitatively and quantitatively. However, when applied to trajectory inference-specific tasks, such as fate prediction and pseudotime inference, it significantly underperformed compared to T-time (e.g. fate TV RF-PHATE: $0.264\pm0.072$, T-time: $0.170\pm0.071$; $p=1e^{-7}$). This performance gap may be explained by limited flexibility in balancing clonal and transcriptional information.

Finally, we evaluated the impact of clonal EMD by comparing it against simpler strategies for deriving $P_{\text{AIR}}^*$. Specifically, we replaced clonal EMD with Jaccard-based clonal overlap between landmarks for T-time, and substituted clonal EMD-based clusters with direct clone-level supervision in RF-PHATE. For T-time, using Jaccard-based operators had little impact on topology-related inference tasks, but led to a significant drop in within-branch pseudotime correlation (T-time$_{\text{Jac}}$: $0.161\pm0.097$, T-time: $0.276\pm0.142$; $p=0.0003$), suggesting that encoding transcriptomic similarity into $P_{\text{AIR}}^*$ through clonal EMD mitigates the loss of local transcriptomic information. For RF-PHATE, our custom supervision with clonal EMD-based clusters proved to be necessary for consistent performance across all tasks.

These results emphasize the importance of a fine-grained and modular balance between modalities that our rooted version of Alternating Diffusion provides, which allows the RNA–AIR weighting to be adapted to dataset-specific characteristics.

### 3.3 T-TIME IMPROVES TRAJECTORY INFERENCE ACROSS BIOLOGY-INSPIRED SCENARIOS

Lastly, we evaluated additional biologically motivated scenarios in which RNA-only methods can fall short, such as converging/cross-converging fates, cyclic topologies, transcriptional noise, or rapidly transitioning trajectories between stable states that create sparsely connected topologies (Wagner & Klein, 2020) (Appendix A.1.5).

As expected, RNA-only methods exhibited the anticipated failure modes, while T-time consistently recovered cellular representations reflective of the ground truth topology across scenarios (Fig. 4a). T-time improvements were particularly notable in fate predictions tasks for the cross-convergence tree and the gaped tree, as well as for pseudotime inference in the cyclic tree (Fig. 4b). In the gaped tree, disconnected components in the underlying graph made CellRank and Palantir fail three and two out of five experiments, respectively. T-time 0/5 had an average fate TV of $0.178\pm0.051$ compared to $0.115\pm0.030$ for T-time. For pseudotime inference in the cyclic tree, T-time maintained a correlation of $0.854\pm0.033$ with the ground truth, while T-time 0/5, the best RNA method, had an average correlation of only $0.693\pm0.069$ ($p=0.002$). Like in the previous experiment, T-time did not provide substantial improvements in pseudotime ordering in the converging tree, nor fate inference in noisy scenarios relative to T-time 0/5. However, although differences were generally non-significant, partially due to the small sample size in this experiment, T-time had the best median performance in the six proposed tasks, compared to both RNA-only and multimodal approaches, corroborating that T-time is the best approach for clonally constrained trajectory inference among the evaluated ones.

This analysis confirmed that RNA-only methods fail in the presence of transcriptomic convergence or missing intermediate states, and that T-time can be used to recover faithful trajectories.

## 4 CONCLUSIONS

We introduced **T-time**, a principled framework for integrating endogenous lineage information with transcriptomic data in snapshot scRNA-seq. By combining transcriptomic and clonal similarity into a lineage-constrained transition map through step-size-controlled alternating diffusion, T-time infers cellular representations, terminal fates, and pseudotime that reflect lineage relationships while preserving local transcriptomic structure. Through synthetic experiments, we provided preliminary evidence that integrating clonal information can resolve the failures of RNA-only approaches in the presence of transcriptomic convergence or missing intermediate states, providing a path to more accurate, lineage-aware single-cell analyses.

While clonally constrained trajectories improved the recovery of the differentiation topology and fate estimation, they offered limited benefits in datasets with high transcriptomic noise, suggesting

that while injecting transcriptomic information into the clonal distance metric is useful to preserve local similarities, it can be detrimental in noisy settings. Future work should therefore systematically investigate the impact of transcriptomic noise, as well as clonal distribution and sparsity, to guide model configuration. Importantly, empirical evaluation with real data is required to thoroughly validate the transferability of our method. While motivated by T-cell differentiation, T-time generalizes to other endogenous lineage barcodes, offering a broadly applicable strategy for constraining trajectory inference with lineage information.

MEANINGFULNESS STATEMENT

We consider representations meaningful when they reflect the system's underlying generative process, enabling interpretations aligned with true biological causality. In trajectory inference, this means that proximity in the learned embedding should correspond to developmental relatedness, not just transcriptional similarity. We achieve this by constraining cell-state embeddings with endogenous lineage information. Incorporating clonal ancestry into diffusion-based representations biases the embeddings toward biologically plausible trajectories that respect lineage continuity, producing representations that are more faithful to the true developmental process and more informative for downstream analyses.

ACKNOWLEDGMENTS

This work is partially funded by the Helmholtz International Lab Causal Cell Dynamics (InterLabs-0029) - Grant support from the Initiative and Networking Fund of the Hermann von Helmholtz-Association Deutscher Forschungszentren e.V. I.B. is supported by the Helmholtz Association under the joint research school "Munich School for Data Science—MUDS", M.L. by the FRQNT Doctoral research scholarship and G.W by a Canada CIFAR AI chair (via Mila), a Humboldt research fellowship, the NSERC Discovery grant 03267, and the NSF grant DMS-2327211. The authors declare no competing interests. The content provided here is solely the responsibility of the authors and does not necessarily represent the views of the funding agencies.

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

# A   APPENDIX

## A.1   SUPPLEMENTARY TEXT

### A.1.1   DETAILS ON OPTIMAL TRANSPORT (OT)

We describe here how we use optimal transport (OT) Peyré & Cuturi (2018) to define clonal distances and landmark-to-landmark transitions. For computational efficiency, we fit a small PHATE operator Moon et al. (2019) with $m' = n/10$ landmarks, which we view as coarse clusters. Then, we represent each clone as a distribution over these clusters. For each pair of clones $k$ and $l$, we then compute OT using informational distance between clusters, derived from the PHATE algorithm, as the cost:

$$(D_{\text{AIR}})_{kl} = \min_{\substack{T \in \mathbb{R}^{m' \times m'} \\ T\mathbf{1}=q_k \\ T^T\mathbf{1}=q_l}} \left[ \sum_{1 \leq i,j \leq m'} T_{ij} C_{ij} \right]$$

where $C_{ij}$ contains the diffusion potential between cluster $i$ and cluster $j$, $T$ is a matrix representing a transport plan between clusters subject to some marginal constraints, $q_k$ (resp. $q_l$) is the distribution of cells from clone $k$ (resp. $l$) over the clusters, and $\mathbf{1}$ is a column vector of ones.

Then, using the $m$ landmarks defined in Section 2.2, which are also clusters of cells, we can compute landmark-to-landmark transitions based on OT, with the clonal distances as cost. For landmarks $k, l$, this gives:

$$(P_{\text{AIR}}^*)_{kl} = \min_{\substack{T \in \mathbb{R}^{m \times m} \\ T\mathbf{1}=p_k \\ T^T\mathbf{1}=p_l}} \left[ \sum_{1 \leq i,j \leq m} T_{ij} (D_{\text{AIR}})_{ij} \right]$$

where $T$ is a matrix representing a transport plan between landmarks and $p_k$ (resp. $p_l$) is the distribution of cells from landmark $k$ (resp. $l$) over the clones.

### A.1.2   CLONAL DIFFUSION OPERATOR BASED ON JACCARD SIMILARITY

Instead of using EMD to define a clonal diffusion operator, landmark-to-landmark clonal similarities can be obtained by comparing clonal frequencies across landmarks with a clonal Jaccard index, since each landmark is a cluster of samples. We can thus redefine the $m \times m$ $P_{\text{AIR}}^*$ to be

$$P_{\text{AIR}(i,j)}^* = \frac{|C(l_i) \cap C(l_j)|}{|C(l_i) \cup C(l_j)|}$$

where $C(l) = \{C(x_i)\}_{i \in I(l)}$ and $I(l)$ denotes the set of indices that make up the landmark $l$.

### A.1.3   ALTERNATIVE APPROACHES TO MULTI-MODAL TRANSITION MAPS

**Alternating diffusion** (AD) (Talmon & Wu, 2019) is a method based on diffusion geometry that extracts only the structure shared across sensors while filtering out modality-specific variability by composing diffusion operators. The alternating diffusion is computed from modality-specific diffusion operators by simple matrix multiplication, in our case, giving

$$P_{\text{alt}} = P_{\text{RNA}} P_{\text{AIR}}$$

This operation can be interpreted as alternating steps of a random walk in either modality, such that only the shared information remains. $P_{\text{alt}}$ is then raised to some power $t$, and is typically transformed into diffusion distances that are used for representation learning.

**Integrated diffusion** (ID) (Kuchroo et al., 2021) improves on alternating diffusion by selecting an exponent for each modality separately, using spectral entropy. This method allows for a denoising level tuned to each modality. The integrated diffusion operator is computed as

$$P_{\text{int}} = P_{\text{RNA}}^{*}{}^{t_{\text{RNA}}} P_{\text{AIR}}^{*}{}^{t_{\text{AIR}}}$$

Another approach we explored is supervised representation learning methods, which view AIR information as a label that guides the output representation. This is appropriate for fusing a quantitative

modality (RNA-seq) with a categorical one (AIR). To this end, we apply **RF-PHATE** (Rhodes et al., 2021). RF-PHATE trains a random forest (RF) on the gene expression data, using the AIR labels as the prediction target. Then, RF-GAP (Rhodes et al., 2023) proximities between samples $x, y$ can be computed by counting the number of trees in which the two samples are classified in the same terminal leaf, with careful consideration for the size of the leaf and the multiplicity of $y$ in the training set of this tree. Intuitively, the proximities capture whether two samples are similar in the features that are relevant for classifying the clones. Row-normalizing the proximities gives $P_{\text{RF-PHATE}}$, which can be used in our pipeline like any other diffusion operator.

In practice, we use *clonal clusters* as prediction targets rather than the AIR labels directly, reducing the number of clonal labels from a large $n_{\text{clones}}$ (often $> 1,000$) to a much smaller $m_{\text{clones}}$. The clonal clusters are formed from hierarchical clustering on $D_{\text{AIR}}$ (see Section 2.2) to group clones with similar differentiation profiles, We avoid excluding the very prevalent non-expanded clones (clones containing fewer than 3 cells) by clustering them separately into a few groups using Leiden clustering.

### A.1.4 Joint simulation of transcriptome and clonal information

We jointly simulate gene expression and clonal labels by extending PROSSTT (Papadopoulos et al., 2019) to include clonal information through a known cellular differentiation tree. PROSSTT generates dynamic single-cell datasets representing complex differentiation processes by defining gene program activation along the branches and pseudotime of an input tree. Cells are generated by sampling gene expressions from a binomial or negative binomial distribution at different branches and times based on the corresponding gene program activation, allowing for modulating noise levels.

We extended PROSSTT to include clonal information by defining *lineages* as restricted paths on the tree topology, and *clones* as sets of cells originating from a common ancestor sampled along a single lineage. Clonal behavior is parametrized by clone frequency, expected size, and pseudotime distribution, capturing stereotypical regimes such as frequent naive clones with few and early cells, or rare activated clones with many and late cells. For each clone, cells are sampled from a single lineage according to the clone's defined behavior, alongside their gene expression generated by PROSSTT.

### A.1.5 Simulation scenarios

We evaluate RNA-based methods and compare them to our clonal-informed approach through a series of biologically-inspired situations that can hypothetically lead to the failure of RNA-only methods (Wagner & Klein, 2020).

- **Convergence**, **crossed convergence** and **crossed convergence (big)**: Variants with no branches, two branches or four branches after convergence. *Challenge:* RNA-only methods may fail to correctly order *pseudotime* across converging trajectories or distinguish terminal fates. *Evaluation target:* Accurate *pseudotime* and terminal fate inference under different branching complexities. *Simulation Setup:* 9, 4 and 10 datasets with an average of 1008, 20078 and 3741 cells and 100, 200 and 380 clones, respectively.

- **High noise**: Simulates elevated transcriptomic variability. *Challenge:* High noise can obscure branch connectivity and trajectory structure. *Evaluation target:* Correct reconstruction of tree topology and terminal fate prediction. *Simulation Setup:* 5 datasets with an average of 1,319 cells and 100 clones.

- **Gaps**: Models rapid transitions between stable states with few sampled intermediate cells, reflecting experimental sparsity in immune cell datasets (Suo et al., 2024). *Challenge:* Sparse sampling fragments the trajectory manifold. *Evaluation target:* Ability to link disconnected states and recover continuous *pseudotime*. *Simulation Setup:* 6 datasets with an average of 1,766 cells and 100 clones.

- **Cyclic topology**: Models a scenario where cells rapidly activate from quiescence and return, with minimal intermediate states. *Challenge:* RNA-only methods cannot reliably determine the relative age of activated versus returning cells. *Evaluation target:* Correct temporal ordering along activation–deactivation cycles. *Simulation Setup:* 7 datasets, with an average of 2,188 cells and 200 clones.

## A.2   SUPPLEMENTARY FIGURES

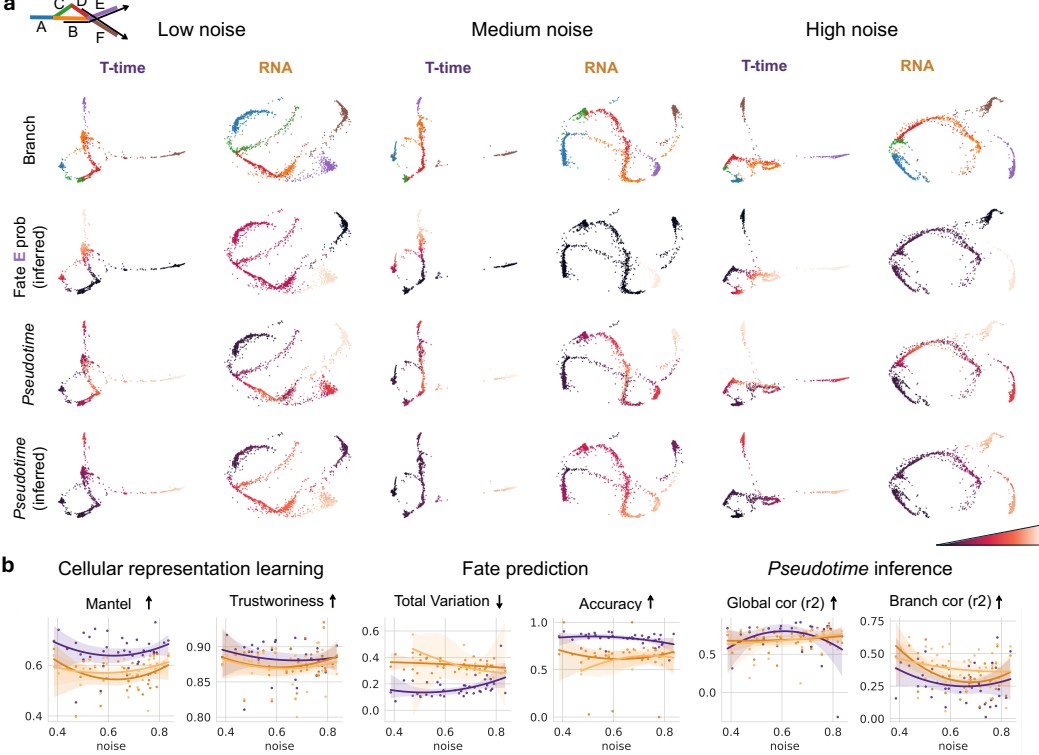

Figure 1: **Extended comparison of T-time to RNA-based models.** **(a)** Representative T-time and PHATE embeddings of a cross-converging tree across three levels of noise, colored by branch, inferred probability to commit to fate E, ground truth pseudotime, and inferred pseudotime. For RNA-based models, the results with the best-performing method are shown in each case. **(b)** T-time quantitative comparison to RNA-based models on cellular representation learning, fate prediction, and pseudotime inference tasks. The y-axis represents performance, and the x-axis noise levels, measured as the average standard deviation in the gene expression from cells belonging to the same branch and pseudotime. Arrows indicate the desired direction of each metric.

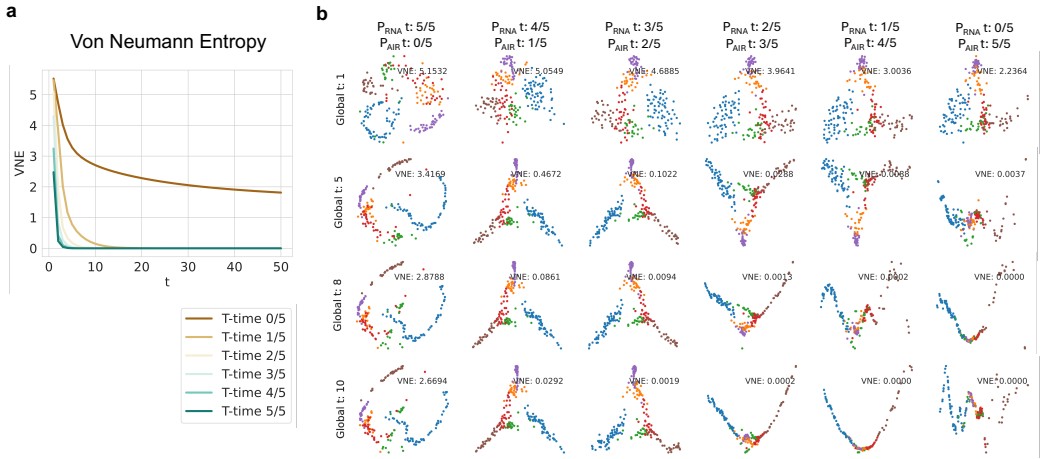

Figure 2: **Impact of $t$ and clonal power on $P_{\text{T-time}}$ entropy and cellular representations**. **(a)** Von Neumann Entropy of $P_{\text{T-time}}$ at various values of $p$. **(b)** T-time embeddings at various values of $p$ and $t$.

