# OpenReview forum: "Multimodal Manifold Learning for Clonally Constrained Trajectory Inference"
_ICLR.cc/2026/Workshop/LMRL — ICLR 2026 Workshop LMRL Poster_

### Official Review · Reviewer_9irh · 2026-02-10
**Solid multimodal representation learning for lineage-constrained dynamics**

**Rating:** 8
**Confidence:** 4

**Review:**

I find this paper to be of good overall quality. While the core idea of incorporating multimodal information into manifold learning is not entirely new, it is executed very well. In particular, Figure 4 presents a convincing and challenging example with crossing branches, where RNA-only geometry is fundamentally ambiguous and the correct structure cannot be recovered without multimodal constraints. This example clearly motivates the method and highlights when and why it is needed. The empirical evaluation is thorough and careful, with appropriate baselines, ablation studies, and statistical reporting (including error bars and significance testing). Overall, the paper is well motivated, technically sound, and clearly presented, and it fits the LMRL workshop well despite not relying on a highly novel conceptual leap.

---

### Official Review · Reviewer_6LxJ · 2026-02-15
**Interesting methodological improvement got trajectory inference with limited validation**

**Rating:** 6
**Confidence:** 3

**Review:**

This work presents T-time, trajectory inference framework that integrates single-cell RNA-seq with Adaptive Immune Receptor (AIR) lineage barcodes to correct developmental paths where transcriptomic similarity fails. Evaluations are presented on synthetic datasets, simulating diverse lineages; e.g., converging, cyclic or gapped topologies.

The paper is well written, and easy to follow--provides the motivation for lineage info, suggests how to use lineage info, aka T-time and presents evaluations. The originality of the paper lies in using endogenous clonal barcodes (AIR); available in dedicated snapshot assays of lymphocytes, as a trajectory constraint, incorporated using solid theoretical tools. The main limitation of the paper, limiting the ability to judge its trus significance are the evaluation. Results are presented on synthetic data, hence applicability, relevance and value for real biological datasets is unclear.

---

### Official Review · Reviewer_MGTH · 2026-02-24
**Review of submission 45**

**Rating:** 8
**Confidence:** 3

**Review:**

Summary

This work introduces T-time, a framework that uses retrospective lineage-tracing to complement snapshot scRNA-seq for trajectory inference. The core idea behind the work is two construct two diffusion operators (transition matrices), one using the lineage information and the other using the scRNA-seq data, then doing a weighted fusion of the two to yield a single diffusion operator. This operator is then used to embed the cells and infer the pseudotime and terminal fate of the cells. The paper evaluates the approach on a synthetic dataset generated from cell trajectories that highlight the failure modes of existing, RNA-only trajectory inference methods.
Strengths
- The problem is well-motivated. scRNA-only trajectory inference has some fundamental limitations, and the use of AIRs to supplement the scRNA-seq snapshots is quite sensible, particularly given the accuracy and availability of these data.
- The experimentation on the synthetic data is extremely thorough, with several ground-truth trajectories, a broad set of baselines, and some ablations as well. This helps the reader easily understand both the strengths and limitations of the method, and motivates many of the design choices the authors made (i.e., the landmarking approach)
- The method shows strong performance in global structure recovery.

Weaknesses
- T-time with the lineage information doesn't uniformly outperform the RNA-only variant, in pseudotime recovery. However, this is not entirely surprising, since recovering pseudotime does not necessarily require differentiating trajectories.
- Even though the experiments are clearly stated to be a proof-of-concept, this work would be substantially stronger if evaluated on real-world data where trajectories are known. In these experiments, the synthetic data is generated expressly to be difficult to model with only scRNA-seq snapshots, so the results are somewhat expected.

Specific comments
- The claim that "No existing computational framework leverages retrospective lineage information to
enhance trajectory inference in standard snapshot datasets, where cells are not explicitly partitioned
between ‘ancestors’ and ‘descendants’ through longitudinal experimental designs" is quite broad and seems to be overstated. For example, a quick search found that PhyloVelo [1] also uses retrospective lineage information to supplement snapshot datasets. The proposed method is still methodologically novel, this claim is just a bit too broad
- The term diffusion operator can be a bit inaccessible to those with less technical background. It may be beneficial to just stick with the term transition matrix (which you do use), since (I believe, though I may be mistaken) they are interchangeable in this setting.
- It might be good to say somewhere how the root and terminal states are adjusted to align with the learned manifold
- If I understand correctly, "step-sized controlled alternating diffusion" might be a bit of a misnomer, since you're not actually alternating between the operators?
Questions
- Is there any advantage to parametrizing the weighting with $p, q \in \mathbb{N}$, as opposed to just having a single $\alpha \in [0,1]$? I understand that you're going for the random-walk intuition, but if you're dividing out by $q$ anyway, it seems redundant?
- By choosing q = 5, you don't have any experiments on T-time 1/2. I'm curious whether there's any difference between T-time 1/2 and vanilla AD? Is there an advantage/disadvantage to alternating?

Recommendation
To me, the only significant limitation of this work is a lack of real-world experiments. Because this is a workshop submission and this is a good proof of concept, I am rating it clear accept.

References
[1]Wang, K., Hou, L., Wang, X., Zhai, X., Lu, Z., Zi, Z., ... & Hu, Z. (2024). PhyloVelo enhances transcriptomic velocity field mapping using monotonically expressed genes. Nature biotechnology, 42(5), 778-789.

---

### Meta-Review · Area_Chair_XqtP · 2026-02-27

**Recommendation:** Accept (Poster)
**Confidence:** 5

**Metareview:**

Accept.

---

### Decision · Program_Chairs · 2026-03-02

**Decision:**

Accept (Poster)

**Comment:**

Please see the meta-review.